# Loss of Astrocytic µ Opioid Receptors Exacerbates Aversion Associated with Morphine Withdrawal in Mice: Role of Mitochondrial Respiration

**DOI:** 10.3390/cells12101412

**Published:** 2023-05-17

**Authors:** Kateryna Murlanova, Yan Jouroukhin, Ksenia Novototskaya-Vlasova, Shovgi Huseynov, Olga Pletnikova, Michael J. Morales, Yun Guan, Atsushi Kamiya, Dwight E. Bergles, David M. Dietz, Mikhail V. Pletnikov

**Affiliations:** 1Department of Physiology and Biophysics, Jacobs School of Medicine and Biomedical Sciences, University at Buffalo, Buffalo, NY 14203, USA; 2Department of Pathology and Anatomical Sciences, Jacobs School of Medicine and Biomedical Sciences, University at Buffalo, Buffalo, NY 14203, USA; 3Department of Pathology, Johns Hopkins University School of Medicine, Baltimore, MD 21205, USA; 4Department of Anesthesiology and Critical Care Medicine, Johns Hopkins University School of Medicine, Baltimore, MD 21205, USA; 5Department of Neurological Surgery, Johns Hopkins University School of Medicine, Baltimore, MD 21205, USA; 6Department of Psychiatry and Behavioral Sciences, Johns Hopkins University School of Medicine, Baltimore, MD 21287, USA; 7Solomon H. Snyder Department of Neuroscience, Johns Hopkins University School of Medicine, Baltimore, MD 21205, USA; 8Department of Pharmacology and Toxicology, Jacobs School of Medicine and Biomedical Sciences, University at Buffalo, Buffalo, NY 14203, USA

**Keywords:** astrocytes, µ opioid receptor, morphine withdrawal, oxidative phosphorylation, reward

## Abstract

Astrocytes express mu/µ opioid receptors, but the function of these receptors remains poorly understood. We evaluated the effects of astrocyte-restricted knockout of µ opioid receptors on reward- and aversion-associated behaviors in mice chronically exposed to morphine. Specifically, one of the floxed alleles of the *Oprm1* gene encoding µ opioid receptor 1 was selectively deleted from brain astrocytes in *Oprm1* inducible conditional knockout (icKO) mice. These mice did not exhibit changes in locomotor activity, anxiety, or novel object recognition, or in their responses to the acute analgesic effects of morphine. *Oprm1* icKO mice displayed increased locomotor activity in response to acute morphine administration but unaltered locomotor sensitization. *Oprm1* icKO mice showed normal morphine-induced conditioned place preference but exhibited stronger conditioned place aversion associated with naloxone-precipitated morphine withdrawal. Notably, elevated conditioned place aversion lasted up to 6 weeks in *Oprm1* icKO mice. Astrocytes isolated from the brains of *Oprm1* icKO mice had unchanged levels of glycolysis but had elevated oxidative phosphorylation. The basal augmentation of oxidative phosphorylation in *Oprm1* icKO mice was further exacerbated by naloxone-precipitated withdrawal from morphine and, similar to that for conditioned place aversion, was still present 6 weeks later. Our findings suggest that µ opioid receptors in astrocytes are linked to oxidative phosphorylation and they contribute to long-term changes associated with opioid withdrawal.

## 1. Introduction

Astrocytes, a major glial cell type in the brain, express opioid receptors. Exposure to an opioid dynamically alters the activity of astrocytes, resulting in adaptive structural and molecular changes that can be observed in vitro and in vivo [1,2,3,4,5,6,7,8,9]. However, it is not clear if astrocytes are active mediators of opioid-induced neuroadaptations.

Astrocytic glycolysis is a source of metabolic support for neuronal activity, but the role of astrocytic mitochondria in neuron–astrocyte interplay is understudied [10]. However, data from recent studies suggest that oxidative phosphorylation (OXPHOS) in astrocytes regulates brain bioenergetics and redox balance and could contribute to behavior [10,11,12,13]. Additionally, there is evidence of a link between µ opioid receptors (MORs) and astrocytic mitochondria in vitro [14,15], but there are no in vivo data implicating astrocytic OXPHOS in the molecular and cellular mechanisms of long-term opiate-induced neuroadaptations.

To assess the potential contribution of astrocytic MOR to the behavioral effects of opioids, we generated mice with astrocyte-specific knockout of the gene that encodes MOR-1, *Oprm1* [16], and then measured the responses of the mice to acute and chronic morphine treatments and morphine withdrawal. We also examined the effects of *Oprm1* deficiency on mitochondrial respiration and glycolysis in brain astrocytes in naïve and morphine-dependent mice. We found that astrocyte-specific knockout of *Oprm1* enhanced naloxone-precipitated conditioned place aversion that persisted for 6 weeks. Furthermore, *Oprm1* knockout produced an elevation in OXPHOS that was further enhanced by naloxone-precipitated morphine withdrawal. Our findings suggest that MORs may be linked to OXPHOS in astrocytes which contribute to long-term neuroadaptation produced by exposure to and/or withdrawal from opiates.

## 2. Materials and Methods

### 2.1. Mice

Mice with inducible astrocyte-specific knockout of *Oprm1* (inducible conditional knockout [icKO]) were generated by crossing *Aldh1l1*::Cre-ER^T2^ mice [17] with *Oprm1^fl/fl^* mice (a generous gift from Dr. Raja [Johns Hopkins University] and Dr. Kieffer [McGill University]) [18]. *Aldh1l1* is highly specific to astrocytes and is expressed in a broader population of astrocytes than other commonly used astrocyte markers in mice [17]. Male mice heterozygous for *Oprm1* icKO (*Aldh1l1*::Cre-ER^T2^x*Oprm1^fl/+^*) were the experimental group; male wild-type, *Aldh1l1*::Cre-ER^T2^, and *Oprm1^fl/fl^, Oprm1^fl/+^* mice were included as controls. All mice were treated with tamoxifen and were on the C57BL/6/J background. Mice were reared by their dams until postnatal day (PND) 21; afterwards, four same-sex mice were housed per cage.

Mice were given standard laboratory chow and water ad libitum and were housed in a temperature- (21 ± 2 °C), humidity- (55 ± 5%), and light-controlled room (reversed 12 h light/12 h dark cycle, light on at 6 p.m., light off at 6 a.m.).

### 2.2. Tamoxifen Injections

At PND 30, all mice were injected intraperitoneally (i.p.) for three consecutive days with 100 mg/kg tamoxifen (T5648, Sigma-Aldrich, St. Louis, MO, USA) dissolved in sunflower seed oil (S5007, Sigma-Aldrich), which induced conditional recombination in astrocytes of *Oprm1* icKO mice. The dose volume was 10 mL/kg body weight.

### 2.3. Morphine and/or Naloxone Injections

Morphine sulfate (1448005, USP, Rockville, MD, USA) dissolved in normal saline was injected i.p. for conditional place preference (10 mg/kg) and conditional place aversion (escalating doses of 20, 40, 60, 80, and 100 mg/kg) tests and subcutaneously (s.c.) for analgesic (5 mg/kg) and sensitization (10 mg/kg) tests. Naloxone (N7758, Sigma-Aldrich) dissolved in normal saline was injected s.c. for CPA test (0.25 mg/kg). For all treatments, the dose volume was 10 mL/kg body weight.

### 2.4. Behavioral Testing

At PND 60 (4 weeks after tamoxifen injections), open field test, elevated plus maze, and novel place recognition test were used to assess locomotor activity, anxiety, and recognition memory, respectively. Morphine-induced sensitization, conditioned place preference, conditioned place aversion, hot plate and Hargreaves test were tested in separate cohorts of *Oprm1* icKO and control mice (Table 1).

Mice were gently handled in the housing room for 1 min over the course of 3–5 days prior to behavioral testing. Mice were transported from the vivarium into the test rooms and were allowed to habituate for at least 30 min before testing. Body weight was monitored throughout the whole experimental period, and no genotype-dependent differences were observed. Mice were randomly and evenly allocated to each experimental group. To perform the group allocation in a blinded manner during data collection, animal preparation and experiments were performed by different investigators. All the experiments were carried out during the dark phase, and the red-light intensity was adjusted to ∼20 lux during the behavioral testing.

#### 2.4.1. Open Field Test

Locomotor activity, average speed and thigmotaxis were evaluated in an open field arena made of clear acrylic (40 × 40 × 30 cm) placed inside a frame containing evenly spaced photocells and receptors making a grid of infrared photobeams from the front to the back and from the left to the right (16 photobeams per axis) (Omnitech Electronics, Inc., Columbus, OH, USA). Any movement made by a mouse within the VersaMax monitor broke through the light beams, thus revealing the mouse’s position in the (X-Y) plane. The mouse was placed in the center of the arena, and activity was recorded for a 30 min session. Data was analyzed using the VersaMax animal activity software (v.4.12-1AFE, AccuScan Instruments, Inc., Columbus, OH, USA). Distance moved and time spent in the outer zones (within 1 cm proximity to the walls of the cage) of the arena was interpreted as thigmotaxis; i.e., wall-hugging behavior.

#### 2.4.2. Elevated Plus Maze

Anxiety-related behavior was assessed via an elevated plus maze. Mice were placed in the center of the maze consisting of two open arms and two closed arms 33 cm in length (San Diego Instruments Inc., San Diego, CA, USA) and allowed to explore the maze for 5 min. The elevated plus maze was raised 0.5 m above the floor. Any-maze Tracking Software (v.6.34, Stoelting, Co., Wood Dale, IL, USA) was used to automatically record and analyze preference to the open arms of the maze (% of time spent in the open arms of the maze). Criterion for arm entries was front two paws and snout inside of the arm.

#### 2.4.3. Novel Place Recognition

The novel place recognition test was performed using a square arena (40 cm × 40 cm × 40 cm) made of white plastic. The test was divided into 3 sessions. During the habituation session (1 day before the training session), mice spent 10 min in the arena to decrease the exploration of the novel environment and anxiety during the training and test sessions. During the training session, mice became habituated to two identical objects placed at fixed locations in the arena. Each mouse began the training session at the arena’s center, and the time spent in each object’s immediate vicinity (within 2 cm) was recorded during a 10 min period, after which the mouse was placed back into the home cage. Long-term recognition memory was tested after 24 h, when mice revisited the arena, in which one of the object’s location was changed. Time spent near the objects was again recorded for 5 min. Preference (%) for object in a novel location was calculated from the time mouse spent in each object’s vicinity by dividing the exploration time of the object in a novel location (TN) by the total time spent by the mouse near objects in novel (TN) and familiar (TF) locations: Preference (%) (novel place) = (TN/(TN + TF)) × 100. Data were recorded and analyzed using the TopScan tracking software (v.2.0, Clever Sys., Inc., Reston, VA, USA).

#### 2.4.4. Morphine-Induced Sensitization

Locomotor activity in response to acute effects of morphine and morphine sensitization were assessed in a separate cohort of *Oprm1* icKO and control mice as previously described [19] (Figure 1). The distance moved by each mouse in the activity box was detected by an infrared motion-sensor system (Omnitech Electronics, Inc., Columbus, OH, USA) fitted outside a transparent plastic cage and analyzed with Versa Max animal activity software (v.4.12-1AFE, AccuScan Instruments, Inc., Columbus, OH, USA).

Each mouse was allowed to habituate to the activity box for 30 min 4 days before the test. On day 1 (test 1), mice were placed in the activity box 30 min before receiving an injection of morphine (10 mg/kg, s.c.) or saline, after which locomotor activity was recorded for 2 h. The mice then received similar injections once per day in home cages for the next 6 days (i.e., chronic daily treatment). No treatment or handling occurred on day 8. Locomotor activity in response to morphine or saline was again recorded for 2 h on day 9 (test 2) 30 min after they were placed in the activity boxes. Sensitization scores were calculated as [(distance traveled on day 9) − (distance on day 1)]/(distance on day 9) × 100.

#### 2.4.5. Conditioned Place Preference

Morphine-induced conditioned place preference (CPP) was tested in a separate cohort of *Oprm1* icKO and control mice according to a previously published protocol [11]. CPP scores were calculated as the percent time spent in the morphine-paired compartment minus the percent time spent in the same compartment on the preconditioning day (Figure 2).

#### 2.4.6. Conditioned Place Aversion

Conditioned place aversion (CPA) was tested in a separate cohort of *Oprm1* icKO and control mice as previously described [11] to evaluate the negative affective states induced by morphine withdrawal [20,21] (Figure 3). Mice were injected with escalating doses of morphine (20, 40, 60, and 80 mg/kg, i.p.) or saline twice a day for 4 days (at 9 a.m. and 5 p.m.). Three hours after the morning injection on day 3, mice were allowed to explore the CPA apparatus for 15 min (preconditioning). Place conditioning occurred 3 h after the morning injection on days 4 (saline, s.c.) and 5 (naloxone 0.25 mg/kg, s.c.). On day 5, mice received a single injection of morphine (100 mg/kg) or saline in their home cages. Place conditioning was tested on day 6, 24 h after the naloxone injection. The CPA score was calculated as the percent time spent in the naloxone-paired compartment minus the percent time spent in the same compartment on the preconditioning day.

#### 2.4.7. Analgesic Effects

Analgesic effects of morphine were tested with a hot plate analgesia meter (Columbus Instruments, Columbus, OH, USA). Additionally, Hargreaves test was conducted using an IITC plantar analgesia meter (IITC Life Science Inc., Woodland Hills, CA, USA) as previously described [11]. For both tests, the maximal stimulus time was 30 s (cutoff time). The procedure was repeated 30 min after injection of morphine (5 mg/kg, s.c.), and the antinociceptive morphine response was calculated as a percentage of maximal possible effect (%MPE) using the formula 100 × (morphine response time − basal response time)/(cutoff time − basal response time).

### 2.5. RNA In Situ Hybridization

In situ hybridization was performed using the “BaseScope” method [22] with a kit supplied by ACD (Newark, CA, USA) and 15 μm fresh-frozen sagittal mouse brain sections obtained from naïve *Oprm1* icKO and control mice (PND 90). Custom “3ZZ” probes were designed by ACD and were targeted to *Oprm1* mRNA (GenBank accession NM_001039652.2), nucleotides 845–976, and *Aldh1l1* mRNA (aldehyde dehydrogenase 1 family, member L1; NM_027406), nucleotides 1256–2112. *Aldh1l1* mRNA hybridization was used as a marker of astrocytes and was detected by precipitation of Fast Green. *Oprm1* mRNA hybridization was also detected by precipitation of Fast Red. All sections were additionally stained with 50% Gill’s hematoxylin 1 (s209, Poly Scientific R&D Corp., Bay Shore, NY, USA) to disclose tissue morphology. Images were acquired using a Leica DM 6B upright microscope with a Leica DFC 7000T color camera and a 20× lens objective (numerical aperture 0.8) and a LED white light source (4500 K) using Leica LAS X acquisition software, v.3.7.3.2324.

Expression was evaluated using ImageJ 1.53c [23]. *Oprm1* expression was assessed by selecting random regions of interest (ROI; defined by a 15-μm circle) in a field that contained at least one green-turquoise punctum indicative of an astrocyte. Pixels containing red *Oprm1* staining were then counted within the ROI. Fifteen ROIs were evaluated within each 312 μm by 234 μm image from various brain regions. Images were uniformly adjusted for brightness, contrast, and color balance.

### 2.6. Biochemical Assays

Brain astrocytes were isolated at PND 150 from naïve *Oprm1* icKO and control mice (for Western blotting and the metabolic flux assay) and from mice that underwent the CPA test (for metabolic flux assay). Whole brains were dissected and placed in Dulbecco’s phosphate-buffered saline containing Ca^2+^, Mg^2+^, D-glucose, and pyruvate (14287080, Thermo Fisher Scientific, Waltham, MA, USA). Astrocytes were isolated according to the “Isolation and cultivation of astrocytes from adult mouse brain” Miltenyi Biotec protocol for the Adult Brain Dissociation kit (130-107-677, Miltenyi Biotec, Bergisch Gladbach, Germany) and anti-ACSA-2 MicroBead kit (130-097-678, Miltenyi Biotec). After magnetic sorting, cells were centrifuged for 5 min at 300× *g* at 4 °C. Pellets were dissolved in cell lysis buffer (9803, Cell Signaling Technology, Danvers, MA, USA) for Western blotting or in AstroMACS medium (130-117-031, Miltenyi Biotec) for subsequent cultivation.

#### 2.6.1. Western Blotting

The expression of MORs by isolated astrocytes was evaluated using a standard Western blotting protocol [11]. The nitrocellulose membrane was incubated overnight at 4 °C with anti-MOR1 mouse monoclonal antibody (1:500, sc-515933, Santa Cruz Biotechnology, Dallas, TX, USA) and then with horseradish peroxidase-conjugated anti-mouse secondary antibody (1:1000, NA931, Cytiva, Marlborough, MA, USA). The optical density of protein bands on each digitized image was normalized to the optical density of the band for β-actin (A5441, Sigma-Aldrich) using ImageJ 1.53c.

#### 2.6.2. Metabolic Flux Assays

Energy metabolism was evaluated in isolated astrocytes using an XF96 Extracellular Flux Analyzer (Agilent, Santa Clara, CA, USA). Specifically, mitochondrial OXPHOS was assessed on the basis of the oxygen consumption rate (OCR), and glycolysis was measured by analyzing the extracellular acidification rate (ECAR).

Isolated astrocytes were plated at a density of 50 × 10^3^ per well in XF96 cell-culture microplates precoated with poly-d-lysine (P6407, Sigma-Aldrich). The cultures were incubated for 4 days in AstroMACS medium prior to the assay. On day 5, the medium was replaced with Seahorse XF medium supplemented with 10 mM D-glucose (G8769, Sigma-Aldrich), 1 mM l-glutamine (25030081, Life Technologies of Thermo Fisher Scientific), and 1 mM sodium pyruvate (11360070, Life Technologies,), and OCR and ECAR were measured using Seahorse XF Cell Mito Stress (103015-100, Agilent) and Glycolysis Stress (103020-100, Agilent) test kits.

OCR was determined at baseline and at each point after adding oligomycin (inhibits ATP synthase [complex V]), carbonyl cyanide-4 (trifluoromethoxy) phenylhydrazone (FCCP; uncoupling agent that collapses the proton gradient and disrupts the mitochondrial membrane potential), and rotenone (inhibits complex I). ECAR was determined at baseline and at each point after adding D-glucose, oligomycin, and 2-deoxy-glucose (2-DG; inhibits glycolysis by competitively binding to glucose hexokinase). The OCR and ECAR measurements were recorded after an 11 min equilibration, with a 1 min mix period and a 1 min wait period. All measurements were in pmol per min and were normalized to total protein concentration in each well.

### 2.7. Statistical Analysis

GraphPad Prism 9.4.1 was used for data analysis and graph generation. All data are presented as means ± SEMs and were analyzed with unpaired Student’s *t* tests or two-way ANOVAs with or without corrections for repeated measures. Differences between the groups were further investigated using Bonferroni post hoc tests. The Bonferroni correction was used for multiple behavioral tests on the same mice. Normal distribution was checked using the Shapiro–Wilk test. Power analysis was conducted using G*Power 3.1.9.6 [24] based on previously published work [11] to determine the sample size. Significance was set at a *p* value of <0.05.

## 3. Results

### 3.1. Decreased Levels of Oprm1 mRNA and Protein in Astrocytes from icKO Mice

MORs are highly expressed in brain astrocytes [25,26] and Aldh1l1-Cre/ERT2 selectively targets brain astrocytes following tamoxifen induction [17,27].

We first validated astrocyte-specific knockout of *Oprm1*. BaseScope in situ hybridization was used to assess regional expression of *Oprm1* mRNA. In the *Oprm1* icKO mouse model, there was an approximately 50% decrease in expression in the medial prefrontal cortex, dentate gyrus and CA1-CA2 areas of the hippocampus, caudate putamen, nucleus accumbens, and the ventral tegmental area (Figure 4A). These regions were selected because they are where activity of the Aldh1l1 promoter is maximal [17] and also in these regions there is high expression of MORs [28]. Accordingly, protein levels of MOR were significantly lower in astrocytes from *Oprm1* icKO mice than in those from controls (Figure 4E,F). Astrocytes were acutely isolated from the whole brain the using magnetic beads technology, optimized to yield pure astrocytes as validated by the expression of astrocyte specific factors [29,30]. An approximately 50% reduction in astrocytic *Oprm1* mRNA and protein MOR expression were consistent with the mouse genotype being heterozygous.

### 3.2. Normal Baseline Behavioral Phenotypes of Oprm1 icKO Mice

There were no genotype-dependent differences in general activity in the open field test, in anxiety-like behaviors in the elevated plus maze test, or in spatial memory in the novel place location recognition test (Figure 5).

### 3.3. Effects of Oprm1 icKO on Drug-Induced Behaviors

#### 3.3.1. Increased Morphine-Induced Locomotor Activity but Not Locomotor Sensitization with *Oprm1* icKO

Morphine administration (10 mg/kg, s.c.) increases locomotor activity in mice [19,31]. However, acute injections of morphine produced significantly greater locomotion in *Oprm1* icKO mice than in the controls (test 1) (Figure 6A,B). Because locomotor activation can be further enhanced after chronic administration of a drug in a process known as locomotor sensitization, or “reverse tolerance” [32,33], we tested for this in our mice. Six days of morphine treatment (10 mg/kg, s.c.) induced locomotor sensitization that was comparable in *Oprm1* icKO and control mice (test 2) (Figure 6C–E).

#### 3.3.2. *Opmr1* icKO Mice Exhibit Normal CPP to Morphine

Morphine similarly produced CPP in *Oprm1* icKO and control mice (tested in a separate cohort of male 10-week-old mice) (Figure 7).

#### 3.3.3. *Oprm1* icKO Mice Exhibit Stronger CPA to Morphine Withdrawal

Another cohort of mice was tested for CPA to naloxone-precipitated morphine withdrawal (Figure 3). Relative to that of control mice, *Oprm1* icKO mice showed greater CPA (i.e., spent less time) in the naloxone-paired compartment 24 h after the naloxone injection (Figure 8A). When extinction of CPA was tested 3 or 6 weeks after naloxone treatment, morphine-treated *Oprm1* icKO mice but not control mice continued to show CPA (Figure 8B,C).

### 3.4. Oprm1 icKO Does Not Affect the Acute Analgesic Effects of Morphine

In separate cohorts of male 10-week-old mice, we evaluated the acute analgesic effects of morphine (5 mg/kg, s.c.) using the hot plate and Hargreaves tests. The effects were similar in *Oprm1* icKO and control mice (Figure 9).

### 3.5. Increased OXPHOS in Oprm1-Deficient Astrocytes

Because in vitro studies suggest a link between opioid receptors and mitochondrial functions in astrocytes [14,15], we examined the effects of *Oprm1* knockout on mitochondrial function using the Seahorse XF Cell Mito Stress test. Astrocytes were isolated from the whole brain of naïve (no opioid treatment) *Oprm1* icKO and control mice. Figure 10A shows OCR of isolated astrocytes in response to compounds used in the mitochondrial stress test over time. Three basal rate measurements were taken prior to injection of pharmacological manipulators of mitochondrial respiratory chain proteins. Following the establishment of a basal OCR reading, oligomycin was applied to inhibit (H^+^) flow through ATP synthase, essentially blocking all ATP-linked oxygen consumption. Maximal respiration was initiated by exposing cells to FCCP, which is a an ionophore that transports H^+^ across the mitochondrial membrane leading to collapse of membrane potential and rapid consumption of oxygen. Rotenone prevented mitochondrial respiration by blocking complex I (NADH:ubiquinone oxidoreductase). Three measurements of OCR were obtained following the injection of each drug. Compared to controls, *Oprm1* icKO astrocytes exhibited significantly higher OCRs, indicating increased mitochondrial OXPHOS (Figure 10A). This was observed for basal respiration and the FCCP-induced maximal respiratory capacity.

No genotype-dependent changes were observed in the ECAR, suggesting comparable glycolytic function between the two groups (Figure 10B). *Oprm1* icKO and control astrocytes exhibited similar basal level of glycolysis at the beginning of the experiment. No difference was observed even after the addition of glucose, indicating the ability of *Oprm1* icKO and control astrocytes to perform increased glycolysis. Upon addition of oligomycin, a chemical that inhibits aerobic respiration causing the cells to rely on glycolysis for energy, the *Oprm1* icKO and control astrocytes showed similar acidification rates, also indicating their ability for increased glycolysis. ECAR, following treatment with 2-DG represented acidification associated with non-glycolytic activity was similar in both groups.

These data show that knockout of *Oprm1* significantly increases mitochondrial OXPHOS in astrocytes. Increased OXPHOS in *Oprm1* icKO astrocytes did not lead to compensatory decrease in glycolysis, as opposed to increased glycolysis in the absence of OXPHOS [34], suggesting a complex interplay between glycolysis and OXPHOS to adapt the mechanisms of energy production.

Because of the strong and long-lasting naloxone-precipitated aversion (up to 6 weeks), we isolated the whole brain astrocytes at PND 150 from *Oprm1* icKO and control mice exposed to CPA. Consistent with the OCR data obtained from astrocytes from naïve *Oprm1* icKO and control mice, a strong genotype effect was observed in basal respiration, ATP production (basal respiration—respiration after oligomycin addition), and maximal respiratory capacity (measured after FCCP addition) (Figure 11). OXPHOS was increased in *Oprm1* icKO and control mice that had been exposed to the escalating doses of morphine followed by naloxone-precipitated withdrawal compared to OXPHOS levels in saline-treated counterparts (Figure 8C and Figure 11). Of note, was the increase in mitochondrial respiration in astrocytes measured from mice 6 weeks after their last exposure to morphine, suggesting that chronic exposure to morphine and pronounced withdrawal increases mitochondrial energy metabolism that persists for weeks.

## 4. Discussion

Our study sheds light on the behavioral and metabolic functions of astrocytic MORs in a model of morphine dependency. Specifically, we showed that astrocyte conditional knockout of the *Oprm1* gene encoding MOR-1 increased locomotion in mice injected with morphine and enhanced and prolonged conditioned associations with morphine withdrawal. Furthermore, *Oprm1* deficiency increased astrocytic OXPHOS and further enhanced mitochondrial respiration long after morphine withdrawal in mice that were chronically treated with the drug. Taken together, our data support a functional in vivo link between astrocytic opioid signaling and OXPHOS.

It is tempting to speculate that MOR-mediated activation of astrocytic OXPHOS contributes to the negative affect behaviors (e.g., aversion as evidenced by CPA) in animals during protracted abstinence from chronic exposure to opiates [35,36,37]. Our previous work on mice with astrocyte-specific deficiency of *Cox10* (the gene encoding for the complex IV accessory protein COX10) similarly demonstrated a prolonged increase in naloxone-precipitated CPA in morphine-dependent *Cox10* icKO mice [11]. Although the exact reason(s) for the similarity in the CPA data and the similar striking difference in OXPHOS data between *Oprm1* and *Cox10* icKO mice remain unknown, it is not inconceivable that astrocyte mitochondrial respiration could play a significant role in both models. Future studies are warranted to address this issue by performing a more unbiased analysis of other potential changes in both mouse models.

The presence of mitochondria in the fine processes of astrocytes [12] raises the possibility that astrocytic OXPHOS supports neuronal activity [13]. Notably, impairment of OXPHOS in astrocytic mitochondria does not appreciably affect astrocytes themselves [34]. Our work supports data from in vitro studies linking MOR activation and mitochondria [14,15]. Nevertheless, the mechanism(s) by which astrocytic OXPHOS influences the behavioral effects of opioids remains unknown. Chronic exposure to opioids impairs glutamate uptake via downregulation of the plasma membrane glutamate transporter GLT-1 [38,39], and plasmalemmal glutamate transporters are in proximity to mitochondria in astrocytes [40,41,42,43] and interact with the Na^+^/K^+^ ATPase [44]. Thus, astrocytic OXPHOS might affect the activity of the Na^+^/K^+^ ATPase, thereby impacting the trafficking of glutamate transporters and glutamate uptake. Indeed, inhibition of mitochondrial function in astrocytes via fluorocitrate increases glutamate excitotoxicity in neuron–astrocyte co-cultures [45]. Additionally, activation of astrocytic MORs increases the release of glutamate by astrocytes in the hippocampus and nucleus accumbens [46]. Interestingly, activation of GLT-1 in the hippocampus via ceftriaxone reduces the signs of morphine withdrawal [47], indicating that glutamate may be a driver of the effects of morphine withdrawal.

Additionally, inflammatory signaling in astrocytes might influence negative emotional states associated with opioid withdrawal because genes involved in neuroinflammatory signaling (i.e., *Tnf*) are upregulated in glial cells in the amygdalae of rats experiencing morphine withdrawal [48].

There are several limitations of the study. First, we only examined male mice to avoid the influence of altered estrogenic responses in female mice as a result of our use of the selective estrogen receptor modulator tamoxifen to induce Cre recombinase [49]. Another reason was to avoid the variability introduced by the cyclical variation of female sex hormones during the estrous cycle which might affect basic and drug-related behaviors [50,51]. However, we plan to evaluate potential sex-dependent outcomes in future experiments. This is a critical issue because there is a growing appreciation of the influence of estrogens on glial function [52,53,54,55] and there are sex-dependent effects in non-contingent and contingent behavioral responses to drugs of abuse, including opioids [56,57]. Notably, pain sensitivity varies between men and women, with women reporting more chronic conditions that cause pain and are more likely to use prescription opioids than men [58].

Another limitation of the present work is that our MOR knockout affected astrocytes throughout the brain; thus, effects mediated from specific brain regions could not be resolved. For example, a recent work by Nam et al. [7] showed that activation of hippocampal astrocytic but not neuronal MORs was necessary for CPP. The selective MOR agonist, [D-Ala2, N-MePhe4, Gly-ol]-enkephalin (DAMGO), was injected either locally into the hippocampal CA1 region of mice or systemically to induce CPP. Manipulation of MOR expression using short hairpin RNA demonstrated that MOR expression in CA1 hippocampal astrocytes was both necessary and sufficient for DAMGO to induce CPP. In contrast to Nam et al. [7], we did not find any difference between control and *Oprm1* astrocyte icKO mice in CPP in our model with brain-wide knockout of the gene in astrocytes; more brain region-specific manipulations will be required to directly address brain region-dependent contributions of astrocyte MOR to the positive and negative valence behaviors. Another possibility for why our results differ from those of Nam et al. is the extent of MOR knockout, i.e., our mice had a 50% reduction in astrocytic MOR expression, consistent with the mouse genotype being heterozygous. It is possible that the remaining 50% of MOR is sufficient for CPP. Complete elimination of astrocyte *Oprm1* expression, preferably in a circuit-related manner, would improve our understanding of the biology and behavioral roles of this gene. Future studies will utilize Cre-responsive viral reagents delivered postnatally within mesolimbic system to avoid other limitations associated with *Aldh1l1*::Cre-ER^T2^ -mediated recombination, e.g., promotor activity in peripheral organs and *Aldh1l1* expression in neural stem cells [59,60] that were not assessed in this work.

Future research should also assess the roles of astrocytic MORs in opioid self-administration and/or in seeking abstinence or extinction to determine if astrocytic MORs and bioenergetics are potential targets for treatments. Finally, there are three other types of opioid receptors [61], and thus additional work is needed to determine their roles to provide a more comprehensive picture of the molecular changes produced by the deletion of *Oprm1* in astrocytes.

In conclusion, our findings for the first time show that astrocytic MORs influence behavioral responses to morphine and to morphine withdrawal. These data also suggest a potential link between astrocytic MORs and mitochondrial OXPHOS that could be engaged by drugs of abuse to contribute to long-term neuroadaptations. Further research in this direction may yield new strategies for manipulating endogenous opioid signaling to combat opioid dependence.

## Figures and Tables

**Figure 1 cells-12-01412-f001:**
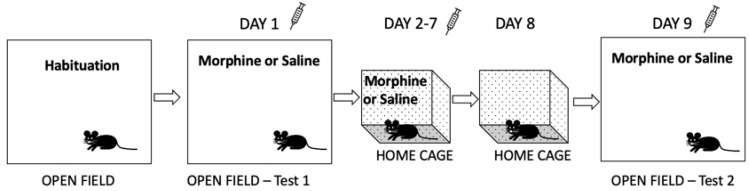
Schematic of the morphine sensitization procedure. Mice were individually habituated to the activity boxes for 30 min, 4 days before the test. On day 1 (test 1), mice were exposed to the activity box for 30 min, followed by a subcutaneous injection of morphine (10 mg/kg) or saline, and locomotor activity was recorded for 2 h. The same mice were administered morphine (10 mg/kg) or saline, respectively, once per day in their home cages for the next 6 days (total of 7 days of chronic daily morphine or saline treatment). On day 8, mice were not handled or treated. On day 9 (test 2), mice were treated the same as on day 1 (test 1), i.e., after a 30 min habituation period, morphine (10 mg/kg) or saline was administered, and activity was monitored for 2 h.

**Figure 2 cells-12-01412-f002:**
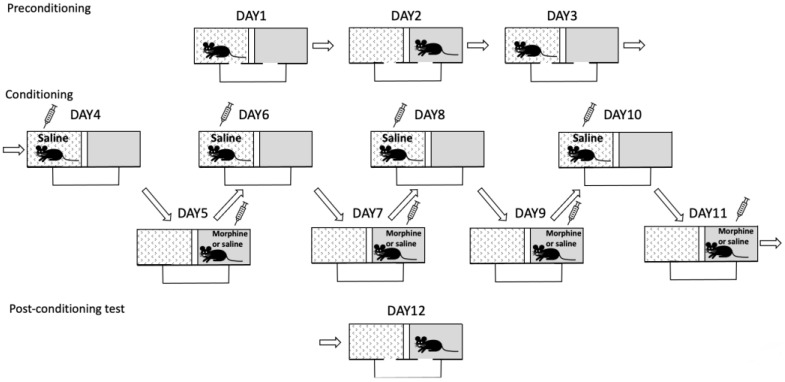
Schematic of the conditioned place preference (CPP) procedure. Preconditioning (habituation): The initial preference of a mouse for one of the compartments was tested. The mouse was placed in the bridge connecting two compartments with different floor textures and different wall patterns. The mouse was allowed to freely explore both compartments for 15 min daily for 3 days. The compartment visited for less time on day 3 was designated the less preferred side and was paired with morphine during the conditioning session. Conditioning: Over the course of 8 days, the mouse was given a morphine or saline injection on alternate days (i.e., 4 days for each treatment) and kept in one of the compartments for 30 min. The mouse was confined to the less preferred side immediately after morphine injection (10 mg/kg; intraperitoneally) and was confined to the more preferred side immediately after saline injection. Half of the control and *Oprm1* icKO mice were given morphine/saline combinations in the opposite compartments, the other half received saline in both compartments. **Testing:** One day (24 h) after the last conditioning session, mice were tested for CPP by placing them in the bridge with open access to both compartments and recording the amount of time spent in each compartment over a 15 min period.

**Figure 3 cells-12-01412-f003:**
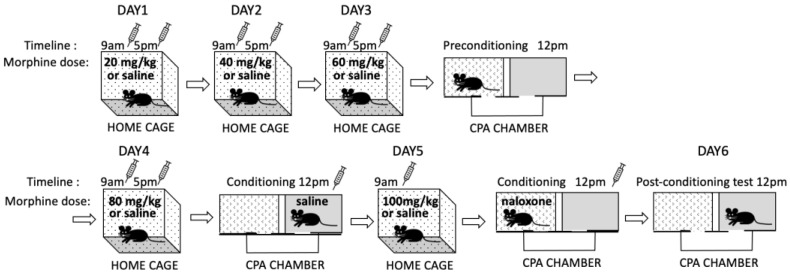
Schematic of the conditioned place aversion (CPA) procedure. A mouse received escalating doses of morphine (20, 40, 60, and 80 mg/kg; intraperitoneally) or saline twice a day for 4 days (at 9 a.m. and 5 p.m.). On day 5, the mouse received a single injection of morphine (100 mg/kg) or saline in the home cage. Three hours after the morning injection on day 3, the mouse was allowed to explore the CPA apparatus for 15 min (preconditioning). Place conditioning was performed on days 4 (saline) and 5 (naloxone 0.25 mg/kg; subcutaneously) 3 h after the morning injection. Place conditioning was tested on day 6, 24 h after the naloxone injection.

**Figure 4 cells-12-01412-f004:**
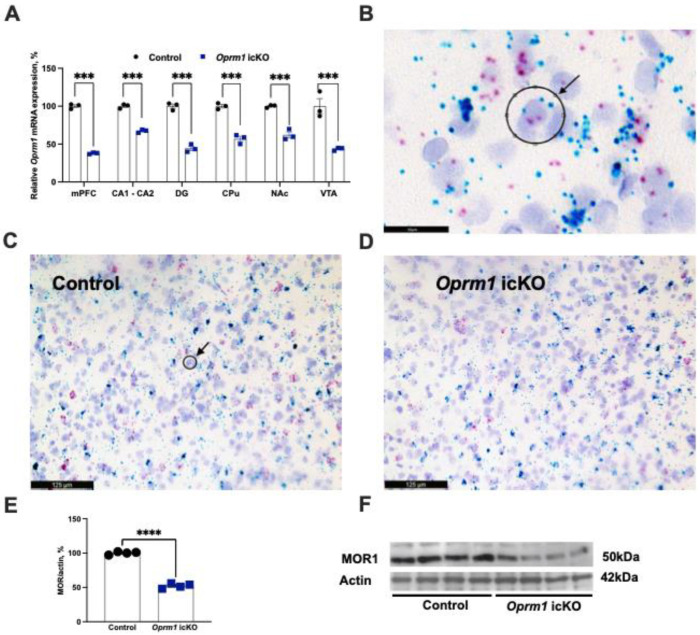
Decreased expression of *Oprm1* mRNA and protein with icKO. (**A**) Quantitative analysis of *Oprm1* mRNA expression in the medial prefrontal cortex (mPFC), CA1-CA2 areas and dentate gyrus (DG) of the hippocampus, caudate putamen (CPu), nucleus accumbens (NAc), and the ventral tegmental area (VTA). Two-way ANOVA revealed a significant effect of genotype [F (1, 24) = 519.3, *p* < 0.001]. A Bonferroni post hoc test showed that *Oprm1* icKO mice expressed significantly less *Oprm1* mRNA in all brain regions of interest. *n* = 3 mice per group; 2 sections per mouse; 15 regions of interest (ROI) per data point. ROI was defined by a 15 μm circle over a field that contained at least one green-turquoise (Fast Green; labeling *Aldh1* mRNA) punctum within an astrocyte, as seen in B. (**B**) Co-localization of *Oprm1* mRNA expression (red dots) surrounding randomly selected *Aldh1l1* mRNA expression loci (green-turquoise dots). The radius of the black circle (ROI) is 15 μm (707 μm^2^). Pixels for red (Fast Red) *Oprm1* mRNA staining were counted within the ROI. (**C**,**D**) Representative images of sections counterstained with 50% Gill’s hematoxylin 1 to disclose tissue morphology, i.e., the cell nuclei (purple). The black circle denotes the ROI shown in B. (**E**,**F**) Quantitative analysis of Western blots revealed significantly decreased expression of astrocytic MOR1 in *Oprm1* icKO mice (Student’s *t* test; *t* = 23.22, *p* < 0.0001). *n* = 4 mice per group; Error bars indicate SEM. *** *p* < 0.001, **** *p* < 0.0001.

**Figure 5 cells-12-01412-f005:**
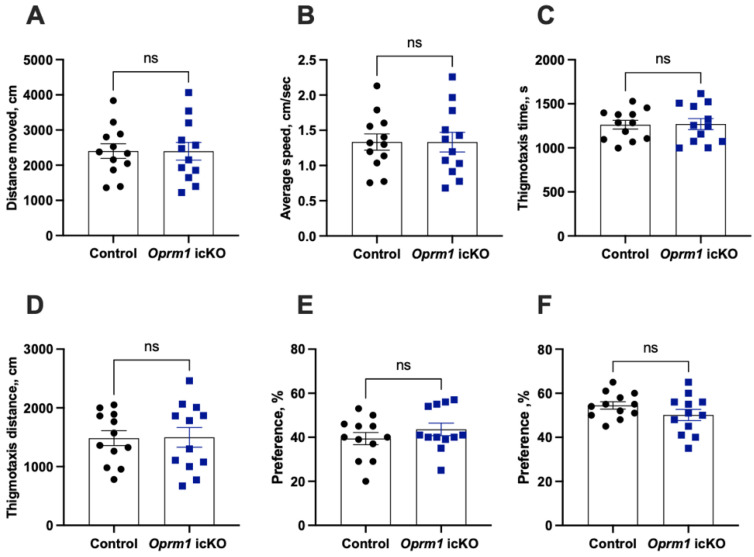
Locomotor activity, anxiety levels, and spatial memory in *Oprm1 i*cKO and control mice. There were no genotype differences in the distance traveled (Student’s *t* test; *t* = 0.009, *p* = 0.9928) (**A**), average speed (Student’s *t* test; *t* = 0.009, *p* = 0.9928) (**B**), thigmotaxis time (Student’s *t* test; *t* = 0.087, *p* = 0.9313) (**C**), thigmotaxis distance (Student’s *t* test; *t* = 0.068, *p* = 0.9463) (**D**) in the open field test, the preference for open arms in the elevated plus maze (Student’s *t* test; *t* = 1.05, *p* = 0.303) (**E**), or the preference for a novel place in the novel place recognition test (Student’s *t* test; *t* = 1.42, *p* = 0.16) (**F**). *n* = 12 mice per group. Error bars indicate SEM.

**Figure 6 cells-12-01412-f006:**
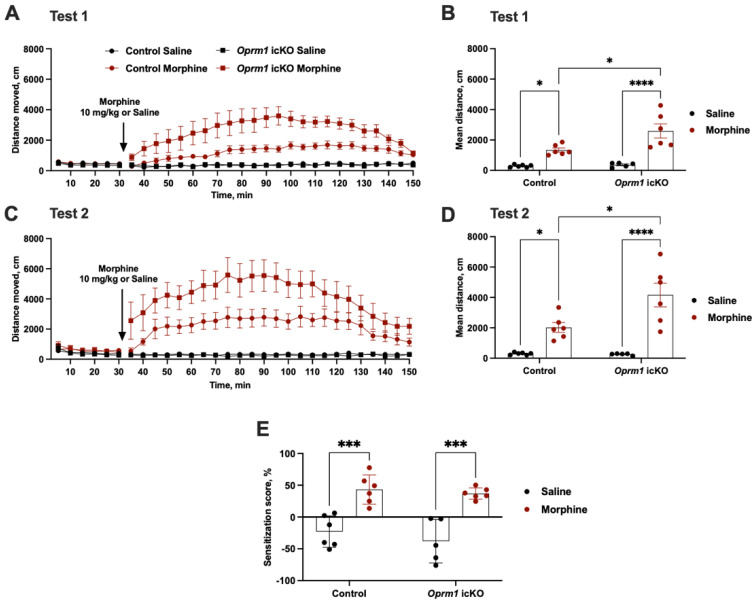
Astrocytic *Oprm1* deficiency leads to increased sensitivity to morphine but not locomotor sensitization to chronic morphine administration. (**A**,**C**) Mice were habituated to the open field (30 min) before locomotor activity in response to morphine (10 mg/kg, s.c.) or saline was assessed. Each data point represents the mean (±SEM) of the distance traveled during a 5 min interval. Both *Oprm1* icKO and control mice exhibited increased locomotion upon morphine injection, but the locomotor-stimulating effect of morphine was more pronounced in *Oprm1* icKO mice. (**B**,**D**) Mean distance traveled after morphine injection. Two-way AVOVA revealed significant effects of genotype [test 1: F (1, 19) = 5.34, *p* = 0.03; test 2: F (1, 19) = 5.59, *p* = 0.02] and morphine [test 1: F (1, 19) = 41.12, *p* < 0.0001; test 2: F (1, 19) = 40.22, *p* < 0.0001] and a genotype by morphine interaction [test 1: F (1, 19) = 5.34, *p* = 0.032; test 2: F (1, 19) = 6.04, *p* = 0.02] on distance traveled. (**E**) Sensitization scores did not differ between *Oprm1* icKO and control mice. Statistical significance between groups was assessed using two-way ANOVA and a Bonferroni post hoc test. *n* = 5–6 per group. Error bars indicate SEM. * *p* < 0.05, *** *p* < 0.001, **** *p* < 0.0001.

**Figure 7 cells-12-01412-f007:**
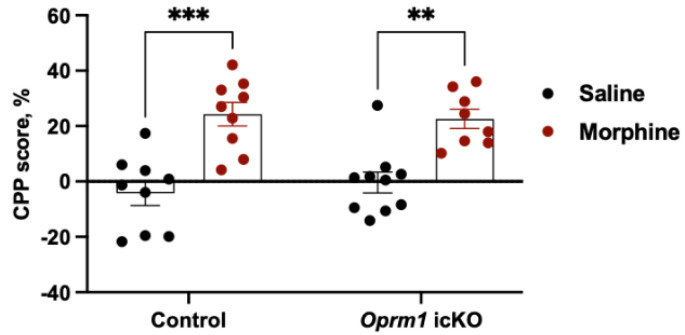
Conditioned place preference (CPP) in *Oprm1* icKO and control mice. Two-way ANOVA revealed a significant effect of morphine [F (1, 32) = 40.24, *p* < 0.0001]. No genotype effect was observed [F (1, 32) = 0.06, *p* = 0.7934]. A Bonferroni post hoc test showed that both morphine-treated control and *Oprm1* icKO mice exhibited significantly increased CPP scores compared to the respective saline-treated groups. *n* = 8–10 per group. Error bars indicate SEM. ** *p* < 0.01, *** *p* < 0.001.

**Figure 8 cells-12-01412-f008:**
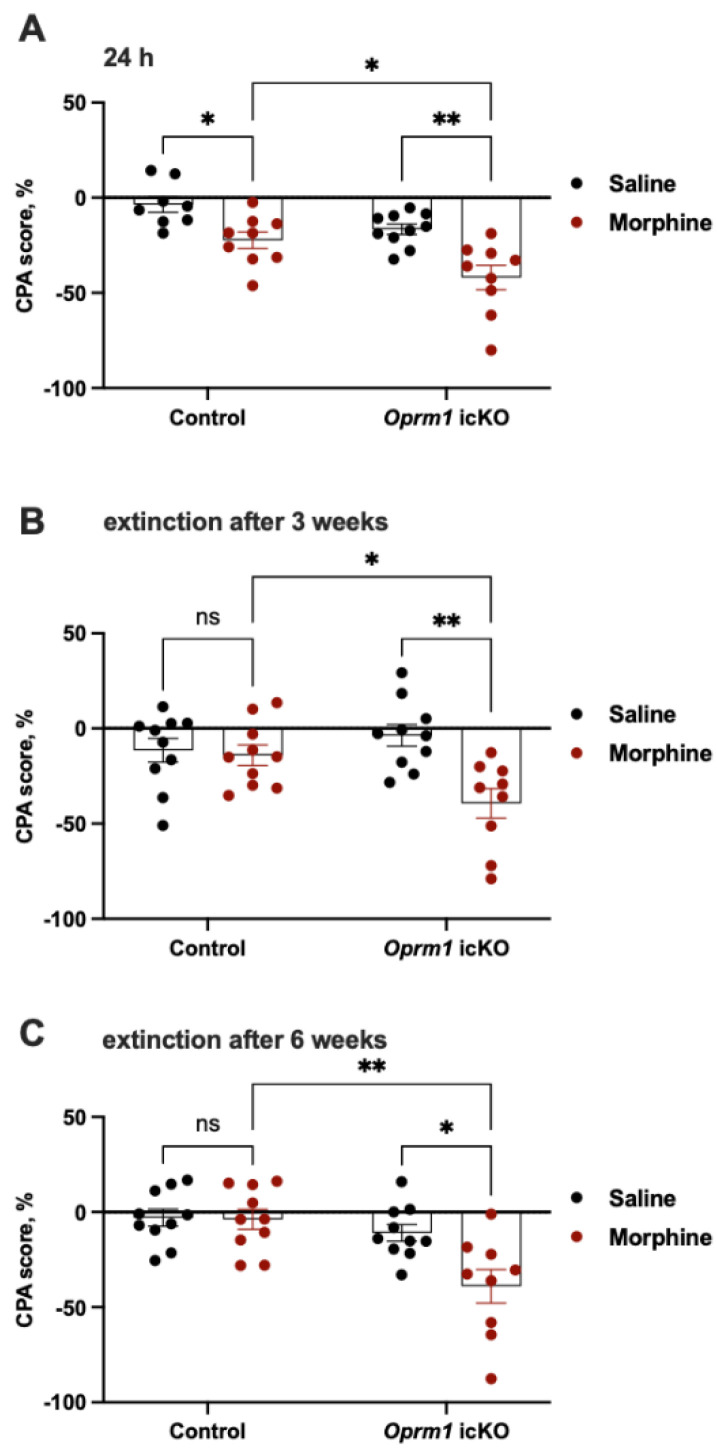
*Oprm1* icKO mice showed an exacerbated CPA and reduced CPA extinction in response to morphine withdrawal. (**A**) Morphine-dependent *Oprm1* icKO mice had stronger aversion (i.e., more negative CPA score) than control mice. Two-way ANOVA revealed significant effects of morphine [F (1, 32) = 23.46, *p* < 0.0001] and genotype [F (1, 32) = 12.81, *p* = 0.001]. A Bonferroni post hoc test also showed that both morphine-treated control and *Oprm1* icKO mice had significantly more negative CPA scores than saline-treated groups. Extinction of CPA was attenuated in morphine-dependent *Oprm1* icKO mice 3 (**B**) and 6 (**C**) weeks after naloxone-precipitated withdrawal. Two-way ANOVA of the CPA data revealed significant effects of morphine [extinction after 3 weeks: F (1, 35) = 9.29, *p* = 0.004; extinction after 6 weeks: F (1, 35) = 6.10, *p* = 0.018], genotype [extinction after 3 weeks: F (1, 35) = 4.70, *p* = 0.03; extinction after 6 weeks: F (1, 35) = 13.55, *p* = 0.0008] and a genotype by morphine interaction [extinction after 3 weeks: F (1, 35) = 6.96, *p* = 0.012; extinction after 6 weeks: F (1, 35) = 5.36, *p* = 0.026]. A Bonferroni post hoc test also showed that morphine-treated *Oprm1* icKO mice had significantly more negative CPA scores than saline-treated *Oprm1* icKO mice. *n* = 8–10 per group. Error bars indicate SEM. * *p* < 0.05, ** *p* < 0.01.

**Figure 9 cells-12-01412-f009:**
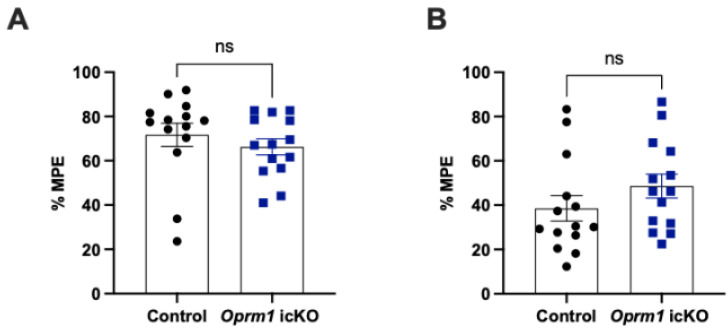
Morphine-induced analgesia in *Oprm1 i*cKO and control mice. %MPE (percentage of maximal possible effect) = 100 × (drug response time − basal response time)/(cutoff time − basal response time). There were no differences between the genotypes in the hot plate test (Student’s *t* test; *t* = 0.84, *p* = 0.407) (**A**) or the Hargreaves test (Student’s *t* test; *t* = 1.27, *p* = 0.214) (**B**). *n* = 14 mice per group. Error bars indicate SEM.

**Figure 10 cells-12-01412-f010:**
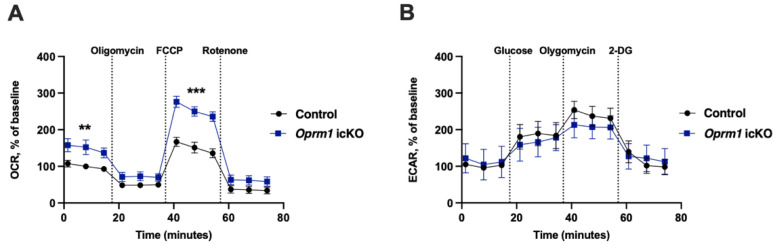
*Oprm1* deficiency increases mitochondrial respiration in astrocytes. (**A**) OCR for control and *Oprm1* icKO astrocytes. Two-way repeated measure ANOVA revealed a significant effect of *Oprm1* icKO for basal respiration [F (1, 9) = 6.04, *p* = 0.036] and maximal respiratory capacity following FCCP injection [F (1, 9) = 32.35, *p* = 0.0003]. Bonferroni post hoc tests showed that OCR at basal respiration and at maximal respiration (measured after FCCP addition) were significantly higher in *Oprm1* icKO astrocytes than in control cells. (**B**) ECAR for control and *Oprm1* icKO astrocytes. Two-way repeated measure ANOVA revealed no significant effects. OCR and ECAR were normalized to the level of basal respiration in control astrocytes. *n* = 5 independent cultures for control astrocytes; *n* = 6 independent cultures for *Oprm1* icKO astrocytes. Each culture was assayed in duplicates. Error bars indicate SEM. ** *p* < 0.01, *** *p* < 0.001.

**Figure 11 cells-12-01412-f011:**
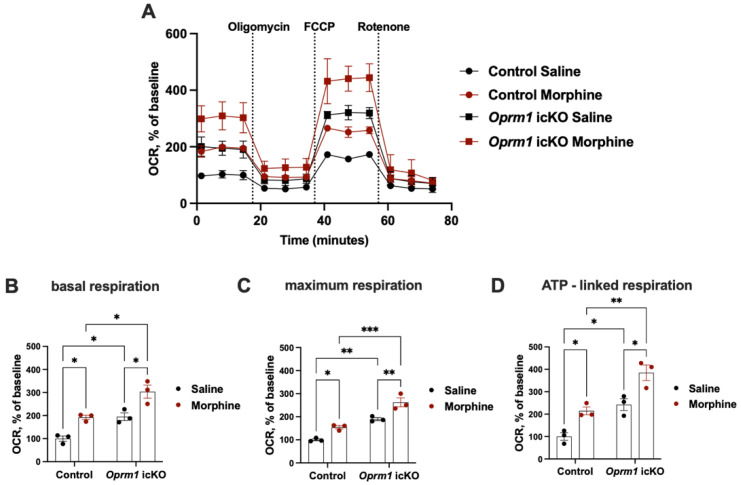
Astrocytic *Oprm1* deficiency leads to long-term increase in OXPHOS in mice exposed to naloxone-precipitated morphine withdrawal. (**A**) OCR in astrocytes isolated from control and *Oprm1* icKO mice 6 weeks after last morphine (or saline) exposure. All mice were exposed to naloxone. The increase in astrocytic OXPHOS resulting from exposure to and withdrawal from morphine was more pronounced in the *Oprm1* icKO group. Basal (**B**), maximum (after FCCP injection; **C**), and ATP-linked (basal respiration—respiration after oligomycin addition; **D**) respiration in astrocytes isolated from morphine-treated *Oprm1* icKO and control mice exposed to naloxone-induced withdrawal and their saline-treated counterparts. Two-way ANOVA revealed significant effects of morphine exposure/withdrawal [basal: F (1, 8) = 30.82, *p* = 0.0005; maximum: F (1, 8) = 32.80, *p* = 0.0004; ATP-linked: F (1, 8) = 26.81, *p* = 0.0008] and genotype [basal: F (1, 8) = 32.72, *p* = 0.0004; maximum: F (1, 8) = 79.24, *p* < 0.0001; ATP-linked: F (1, 8) = 39.76, *p* = 0.0002] on mitochondrial respiration. Statistical significance between groups was assessed using two-way ANOVA and Bonferroni post hoc tests. OCR was normalized to basal respiration of astrocytes isolated from saline-treated control mice. *n* = 3 independent cultures for control and *Oprm1* icKO astrocytes. Each culture was assayed in duplicate. Error bars indicate SEM. * *p* < 0.05, ** *p* < 0.01, *** *p* < 0.001.

**Table 1 cells-12-01412-t001:**
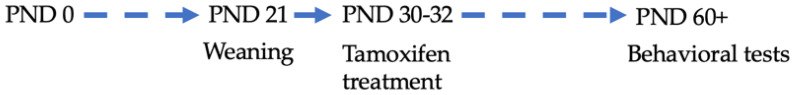
Battery of behavioral tests used in the study.

Cohort	Test	PND
**Cohort 1**	Open field test	60
Elevated plus maze	80
Novel place recognition test	90–92
**Cohort 2**	Morphine-induced sensitization	90–102
**Cohort 3**	Conditioned place preference	60–71
**Cohort 4**	Conditioned place aversion	105–110; 130; 150
**Cohort 5**	Hargreaves test	70
**Cohort 6**	Hot plate test	70

## Data Availability

Data available on request from the authors.

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
