# Peer review of "Loss of Astrocytic µ Opioid Receptors Exacerbates Aversion Associated with Morphine Withdrawal in Mice: Role of Mitochondrial Respiration"

_cells, 2023, doi:10.3390/cells12101412_

Round 1
Reviewer 1 Report
In this study from the Pletnikov lab, the authors extend their very interesting recent findings (Murlanova et al., Glia 2022) surrounding the interplay of morphine treatment and mitochondrial respiration in the control of astrocyte pathogenic activities and behavior. The study is important because it adds to a growing literature of astrocyte functions that mediate a number of physiologically relevant disease-associated processes with potential therapeutic value. The study is well-presented and well-written; the data are clear and the phenotypes are obvious. Importantly, the authors include all of the appropriate controls, hence the conclusions of the study are sound. There are a few points that would strengthen the study:
1. The authors should compare the full Oprm1 cKO to the hets in either the biochemical, metabolic, or behavioral experiments to better understand the strength of the phenotype.
2. There are a few typos throughout and in the abstract the word "preference" is missing on Line 32.
Author Response
We thank the reviewer for the positive evaluation of our work.
There are a few points that would strengthen the study:
- The authors should compare the full Oprm1 cKO to the hets in either the biochemical, metabolic, or behavioral experiments to better understand the strength of the phenotype.
We appreciate the reviewer’s comment that using full KO mice would provide important information. Unfortunately, multiple rounds of generating double-transgenic full Oprm1 cKO mice would take a year, if not longer, to produce sufficient number of mice. This will be an important direction of new studies in the lab. We now indicate this limitation of the current results in the discussion of the manuscript.
- There are a few typos throughout and in the abstract the word "preference" is missing on Line 32.
Thank you. We have corrected all typos.
Reviewer 2 Report
Dear Editor,
The manuscript by Pletnikov et al., entitled “Loss of astrocytic µ opioid receptors exacerbates aversion associated with morphine withdrawal in mice: a role for mitochondrial respiration” has been reviewed for publication in Cells. In this manuscript, the authors demonstrate that deletion of Oprm1 gene in astrocyte increased locomotion and astrocytic OXPHOS in chronic morphine exposure and prolonged these effects in morphine withdrawal. Recently, the authors have published related results in Glia in 2022, and found that astrocyte oxidative phosphorylation may contribute to behaviors associated with greater cognitive load and/or aversive and stressful conditions. Here, the authors further find that µ opioid receptors in astrocytes are linked to oxidative phosphorylation and contribute to long-term changes associated with opioid withdrawal. Although the authors have the new findings, the significance is still limited. As the author mentioned the limitations of this study, there are still many directions to be clarified, such as whether dOR is involved or not. Hence, I suggest the manuscript could not be considered for acceptance for publication in Cells. The manuscript did not receive a high enough priority rating to warrant further consideration.
Author Response
We appreciate the reviewer’s comments. We respectfully think unresolved questions and many new directions will be clarified in the future studies.
Reviewer 3 Report
Murlanova and colleagues have investigated the role of mu opioid receptor 1 (Oprm1) in astrocytes using a conditional knockout of Oprm1 from astrocytes. They performed a series of behavioural and biochemical tests. The main results from behavioural tests are that Oprm1 deficiency in astrocytes increases the morphine-induced increase of locomotion and naloxone-induced conditioned place aversion without having an effect on conditioned place preference. In biochemical tests, Oprm1 deficiency increases oxidative phosphorylation. These are interesting and novel results. The manuscript is well written, and the experiments appear to have been done to a high standard. I have two relatively minor comments.
Nam et al. 2019 reported that activation of astrocytic mu opioid receptor induces conditioned place preference in a conceptually similar experimental design. This seems to be at odds with the results presented in this manuscript. This should be clearly stated, and the potentials reasons should be discussed in detail.
The authors call their approach a ‘knockdown’ in many, but not all, places. I would usually associate ‘knockdown’ with an experimental manipulation targeting expression at the RNA level, e.g., using small interfering RNA (siRNA) or short hairpin RNA (shRNA). In contrast, the cre-lox system modifies the genome, which is typically then called a (cell type specific) conditional knockout. The authors may want to consider this to avoid confusion.
Author Response
We thank the reviewer for the interest in our work and pointing out a paper by Nam et al., 2019, which was cited in the introduction of our manuscript. We now discuss this work in the discussion of our manuscript to address the raised issue.
Another limitation of the present work is that MOR knockout affected astrocytes throughout the brain; thus, effects mediated from specific brain regions could not be resolved. For example, a recent work by Nam et al. showed that activation of hippocampal astrocytic but not neuronal MORs was necessary for CPP. A selective MOR agonist, [D-Ala2, N-MePhe4, Gly-ol]-enkephalin, DAMCO, was injected either locally into the hippocampal CA1 region, or systemically to induce CPP in mice with decreased MOR expression in all brain cells, except for CA1 hippocampal astrocytes [7]. While we did not find any difference between control and Oprm1 astrocyte cKO mice in CPP our model with brain-wide knockout of the gene in astrocytes does not completely rule out a possible involvement of astrocytic MOR in positive reward associated behaviors as more brain region specific manipulation are clearly required to directly address brain region-dependent contributions of astrocyte MOR to the positive and negative valence behaviors.
The authors call their approach a ‘knockdown’ in many, but not all, places. I would usually associate ‘knockdown’ with an experimental manipulation targeting expression at the RNA level, e.g., using small interfering RNA (siRNA) or short hairpin RNA (shRNA). In contrast, the cre-lox system modifies the genome, which is typically then called a (cell type specific) conditional knockout. The authors may want to consider this to avoid confusion.
We thank the reviewer for the comment. We describe our mouse model as conditional knockout (cKO).
Reviewer 4 Report
The goal of this manuscript was to assess the role of the astrocyte mu opioid receptor on behavior and mitochondrial function following morphine exposure. The approach was to use a whole brain knockdown (KD) of the gene that encodes MOR-1, Oprm1 in astrocytes. Using het KD mice, they then assessed the impact of acute and chronic morphine exposure and morphine withdrawal on behavior and astrocyte-specific mitochondrial bioenergetics in naïve and morphine-dependent mice.
This is a well written manuscript that includes appropriate approaches and statistical evaluation. The conclusions support the data presented. The authors demonstrated that astrocyte-specific knockdown of Oprm1 increased conditioned place aversion (CPA) that persisted for a protracted period of time and increased astrocyte-specific oxidative phosphorylation.
The authors address the limitations of their study well and this is appreciated.
Minor Comments:
1. I am not suggesting additional experiments be conducted but it would have been interesting to do a final experiment in which you authors recover mitochondrial function and repeat the behavior in order to determine whether these factors were in-fact causal and not just correlational.
2. Methods section: The 10 mL/kg body weight detail was provided for the morphine and naloxone but the drug amount or range of drug amounts should be included in the methods, i.e., mg/kg?
3. The authors have limited information regarding the open field test, elevated plus maze, and novel place recognition. They refer to two different papers (ref 19, 20) in which (the first) has the absolute minimum detail and the second which lacks details but refers to another paper so you literally have to do a deep dive to find the information. Much more detail should be included in the description of behavioral methodology. Habituation to the room, handling prior to experimental procedures, lux measurements for open field and light dark box, how animal placement in the EPM and open field tests were determined; is the animal considered "in" the open arm when the nose enters? The whole animal? Half the animal? How was this data quantified, what video tracking software etc? Blinding of the experimentor and were the animals randomly assigned to groups, if so how? Do you have weights of these animals throughout the study and if so, are they different? Please give specific ages of the animals on the days of open field, EPM, and age range for the novel object recognition. How long was the delay between AA and AB sessions? Are animals habituated to the object recognition box prior to the AA session, how long are the sessions, and how was time inspecting objects defined, measured, and calculated. These are just a few examples of the type of data that should be included to ensure that a reader can replicate your experiment.
4. Supplementary figures. There is no description of what the bars and error bars are representing. These details are inconsistently included in the figure legends throughout the manuscript. Be consistent with the description across figures.
Figure S5. Please plot the data for time and distance in thigmotaxis in open field. Please include the average speed across the entire open field. Please remember to include how the thigmotaxis zone is defined.
5. Line 281-282 contradicts the data and previous text. “These data show that knockdown of Oprm1 significantly reduces mitochondrial OXPHOS in astrocytes.” This should say increased.
Author Response
We thank the reviewer for the positive evaluation of our work.
Minor Comments:
- I am not suggesting additional experiments be conducted but it would have been interesting to do a final experiment in which you authors recover mitochondrial function and repeat the behavior in order to determine whether these factors were in-fact causal and not just correlational.
We greatly appreciate the comment. Rescuing mitochondrial respiration and behavioral changes will be a future direction of our work.
- Methods section: The 10 mL/kg body weight detail was provided for the morphine and naloxone but the drug amount or range of drug amounts should be included in the methods, i.e., mg/kg?
We apologize for this omission. We now provide the drug amount in the methods’ section.
Morphine sulfate (1448005, USP, Rockville, MD) dissolved in normal saline was injected i.p. for conditional place preference (10 mg/kg) and conditional place aversion (escalating doses of 20, 40, 60, and 80 mg/kg) tests and subcutaneously (s.c.) for analgesic (5 mg/kg) and sensitization (10 mg/kg) tests. Naloxone (N7758, Sigma-Aldrich) dissolved in normal saline was injected s.c. for CPA test (0.25 mg/kg). For all treatments, the dose volume was 10 mL/kg body weight.
- The authors have limited information regarding the open field test, elevated plus maze, and novel place recognition. They refer to two different papers (ref 19, 20) in which (the first) has the absolute minimum detail and the second which lacks details but refers to another paper so you literally have to do a deep dive to find the information. Much more detail should be included in the description of behavioral methodology. Habituation to the room, handling prior to experimental procedures, lux measurements for open field and light dark box, how animal placement in the EPM and open field tests were determined; is the animal considered "in" the open arm when the nose enters? The whole animal? Half the animal? How was this data quantified, what video tracking software etc? Blinding of the experimentor and were the animals randomly assigned to groups, if so how? Do you have weights of these animals throughout the study and if so, are they different? Please give specific ages of the animals on the days of open field, EPM, and age range for the novel object recognition. How long was the delay between AA and AB sessions? Are animals habituated to the object recognition box prior to the AA session, how long are the sessions, and how was time inspecting objects defined, measured, and calculated. These are just a few examples of the type of data that should be included to ensure that a reader can replicate your experiment.
Per the reviewer’s comment, we now describe open filed test, elevated plus maze and novel place recognition tests in greater detail.
- Supplementary figures. There is no description of what the bars and error bars are representing. These details are inconsistently included in the figure legends throughout the manuscript. Be consistent with the description across figures.
We apologize for this omission. We now consistently describe that the bars and error in all figures represent data as means ± SEMs.
Figure S5. Please plot the data for time and distance in thigmotaxis in open field. Please include the average speed across the entire open field. Please remember to include how the thigmotaxis zone is defined.
Per the reviewer’s request we have included new data (thigmotaxis and average speed in open field). Per the editor’s request we have incorporated all the supplementary figures into the main text of the manuscript. The new open field data is now shown in a new figure 6. The definition of thigmotaxis zone is presented in the methods’ section.
- Line 281-282 contradicts the data and previous text. “These data show that knockdown of Oprm1 significantly reduces mitochondrial OXPHOS in astrocytes.” This should say increased.
We thank the reviewer for the comment. We have fixed the mistake.
Round 2
Reviewer 1 Report
No additional comments
Author Response
We thank the reviewer for the interest in our work.
Reviewer 3 Report
The authors have fully addressed my previous comments.
Author Response
We thank the reviewer for the positive evaluation of our work.